# CURRENT ANOMALY DETECTORS ARE ANOMALOUS: ON SEMANTIC TREATMENT OF OOD INPUTS

## ABSTRACT

Machine learning models have achieved impressive performance across different modalities. It is well known that these models are prone to making mistakes on out-of-distribution inputs. OOD detection has, therefore, gained a lot of attention recently. We observe that most existing detectors use the distribution estimated by the training dataset for OOD detection. This can be a serious impediment since faulty OOD detectors can potentially restrict utility of the model. Such detectors, tied to the bias in data collection process, can be impermeable to inputs lying outside the training distribution but with the same semantic information (e.g., class labels) as the training data. We argue that in-distribution should not be tied to just the training distribution but to the distribution of the semantic information contained in the training data. To support our argument, we perform OOD detection on semantic information extracted from the training data of MNIST and COCO datasets, and show that it not only reduces false alarms but also significantly improves detection of OOD inputs with spurious features from training data.

## 1 INTRODUCTION

Machine learning models have achieved remarkable success in accomplishing different tasks across modalities such as image classification (Gkioxari et al., 2015), speech recognition (Hannun et al., 2014), and natural language processing (Majumder et al., 2017). It is however known, that such models are unreliable on samples which are less likely to occur, according to the model's *in-distribution* estimated from its training data (Guo et al., 2017; Hendrycks & Gimpel, 2016). Detection of these *out-of-distribution (OOD)* inputs is important for the deployment of machine learning models in safety-critical domains such as autonomous driving (Bojarski et al., 2016), and medical diagnosis (De Fauw et al., 2018). OOD detection has, therefore, gained a lot of attention recently (Liang et al., 2017; Lee et al., 2018; Hendrycks et al., 2019; Kaur et al., 2021b).

Even though there is sufficient interest in OOD detection, to the best of our knowledge, its unclear what precisely entails an OOD input. Existing detectors estimate a distribution that is tied to the training dataset, and flagging inputs as OOD when the assigned probability according to the estimated distribution is low. The standard drill involves a set of in-distribution inputs drawn from a dataset such as CIFAR10, and detecting those inputs as OOD that are drawn from a different dataset such as SVHN (Hendrycks & Gimpel, 2016; Kaur et al., 2021a; Lee et al., 2018). Such external inputs (from SVHN) would have non-overlapping class labels (from CIFAR10). Ming et al. (2022) show that the existing detectors are unfortunately tied to the sampling bias of the training dataset. This results in low detection on OOD inputs with spurious features such as background, color, etc. from the training data. The authors report low detection performance of existing detectors on two datasets: 1) Birds (Sagawa et al., 2019) with class labels in {waterbirds, landbirds}, and 2) CelebA (Liu et al., 2015) with class labels in {grey hair, non-grey hair}. Table 1 shows these results for OOD images containing water (or land) as spurious feature for waterbirds (or landbirds), and OOD images of bald male with male as spurious feature for grey hair. This means that even though the classifier might be able to generalize better, OOD detectors itself can stifle its utility. On the other hand these detectors can be permeable to *real* OOD inputs which ought to be rejected. With this in mind, we propose to treat *intended distribution* of images as in-distribution. Images containing semantic information relevant to the training classes[1]. Inputs deficient of semantic information

---

[1] We will be using the terms "in-distribution" and "intended distribution" exchangeably in the paper.

Table 1: Low OOD detection by existing detectors on OOD inputs with spurious features from the training data (Ming et al., 2022). **Our Algorithm 2 significantly improves detection on these OOD inputs.**

| Test Set | Baseline | ODIN | Mahala | Energy | Gram | Our |
|---|---|---|---|---|---|---|
| OOD for Birds | 25.32 | 22.75 | 30.65 | 25.78 | 41.75 | **98.97** |
| OOD for CelebA | 16.30 | 18.93 | 21.25 | 28.72 | 18.79 | **74.36** |

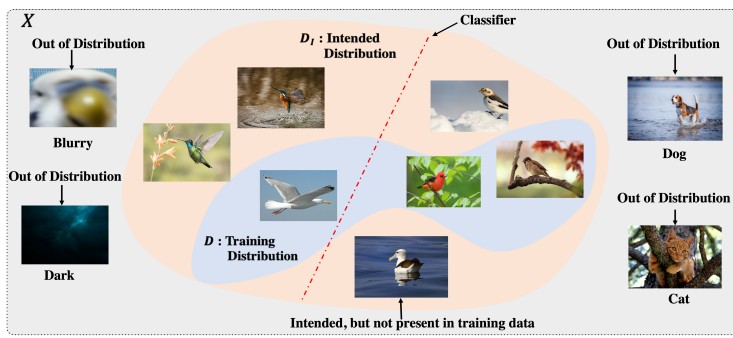

Figure 1: The intended distribution has a much higher variability in terms of the samples it covers, when compared to the training distribution. The classifier trained to classify birds in {sitting birds, flying birds} is expected to generalize well for the intended distribution, which has birds sitting on trees, snow or water. OOD inputs are the ones which are unlikely to occur from the vantage point of the intended distribution $\mathcal{D}_I$.

w.r.t any training class such as bird for the Birds dataset and hair for CelebA need to be detected as OOD.

**Contributions:** We propose two OOD detection algorithms based on distinct ways of estimating the intended distribution. The first algorithm uses expert guided semantic information. When expert guidance is absent, and there is sufficient labeled data, we propose using semantic segmentation network. Table 1 shows that we achieve significant improvement by $57.22\%$ and $45.64\%$ on OOD detection with semantic segmentation networks for Birds and CelebA, respectively. This highlights the drawback of the current approaches.

Our experiments on COCO (Lin et al., 2014) and MNIST (LeCun et al., 1998) datasets show that the existing detectors overfit to the training data for estimating in-distribution, resulting in - 1. false OOD detection on inputs with same (training class) labels but from a separate dataset, and 2. low OOD detection on inputs with classes, absent from the set of training classes. This low detection is due to sensitivity of detectors to the spurious features from the training data. The proposed algorithms not only reduces false alarms significantly but also improves OOD detection ($\geq 20\%$) on inputs with spurious features from training data.

**Related Work.** OOD detection has been extensively studied and detectors with OOD scores based on the difference in statistical, geometrical or topological properties of in-distribution and OOD inputs have been proposed. These detectors can be classified into three categories, supervised (Lee et al., 2018; Kaur et al., 2021a), self-supervised (Hendrycks et al., 2019; Kaur et al., 2022), and unsupervised (Hendrycks & Gimpel, 2016; Liang et al., 2017). Unsupervised approaches can function without an OOD dataset for training the detector, while supervised approaches do. Self-supervised approaches require a self-labeled dataset for training the detector. This dataset is created by applying transformations to the training data and labeling the transformed data with the applied transformation. The proposed OOD detection algorithms in this paper are unsupervised in nature. Ming et al. (2022) show that the existing detectors perform poorly on OOD inputs with spurious features from the training data. They, however, do not propose a solution for fixing the existing detectors.

Domain generalization Zhou et al. (2022) is an active research area where efforts are made for generalizability of machine learning classifier to its classes beyond the training data. It tries to ask the question of whether a classifier trained on the images of birds on trees would work on images of birds on water? Domain-invariant representation learning (Li et al., 2018), training data augmentation with higher variability (Zhou et al., 2020) etc. have been proposed to solve this problem. With the intended distribution of images containing (training) class-specific information for a classifier, we propose inputs that do not contain this information as OOD.

There has been a great interest in making use of semantic segmentation networks in scene understanding problem (Mo et al., 2022), one of the core problems in computer vision with applications e.g. to autonomous driving, video surveillance, and robot perception (Garcia-Garcia et al., 2017). Recently, the use of segmentation networks was proposed to train classifiers with a handful of training examples Mojab et al. (2021). We make use of segmentation networks for OOD detection.

## 2 PROBLEM FORMULATION AND METHODOLOGY

Let $(\mathcal{X}, \mathcal{A}_{\mathcal{X}})$ be the measurable space from which images are sampled. We assume $\mathcal{X}$ is an at most countable subset of a (possibly very high-dimensional) Euclidean space $\mathbb{R}^{h \times w \times 3}$ whose dimension depends on the size of the images. Here, $h, w$ refer to the height and width of the image. Denote by $\Delta(\mathcal{X}, \mathcal{A}_{\mathcal{X}})$ the space of (countably additive) probability measures on $(\mathcal{X}, \mathcal{A}_{\mathcal{X}})$, and consider a candidate distribution $\mathcal{D} \in \Delta(\mathcal{X}, \mathcal{A}_{\mathcal{X}})$.[2] Let then $X_1, \ldots, X_n \sim \mathcal{D}$ iid, denote by $x_1, \ldots, x_n$ the realizations of random variables $X_1, \ldots, X_n$, and call $\mathcal{S} = \{x_1, \ldots, x_n\}$ our training set (also referred to as input sample set).

We assume that the support of $\mathcal{D}$, written supp($\mathcal{D}$), is a proper subset of $\mathcal{X}$, that is, supp($\mathcal{D}$) $\subsetneq$ $\mathcal{X}$. Then, we introduce what we call the *intended distribution*, i.e. a probability measure $\mathcal{D}_I \in \Delta(\mathcal{X}, \mathcal{A}_{\mathcal{X}})$ whose support supp($\mathcal{D}_I$) is a proper superset of supp($\mathcal{D}$). We can write:

$$\text{supp}(\mathcal{D}) \subsetneq \text{supp}(\mathcal{D}_I) \subset \mathcal{X}.$$

Intended distribution $\mathcal{D}_I$ is needed because it assigns non-zero probability to the set of images which are likely to be seen by the classifier in the real world. For instance, in case of standard birds dataset, $\mathcal{D}$ captures images of birds on trees. But, the intended distribution $\mathcal{D}_I$ can refer to birds on trees, water, or snow. We ask the following research question : *Given the input sample set $\mathcal{S}$, can we build an OOD detector that is able to estimate $\mathcal{D}_I$ closely ?*

### 2.1 METHODOLOGY

This is a difficult problem, since it demands generalization from the OOD detectors which are strictly tied to the training set $\mathcal{S}$. For an $\epsilon > 0$, we define the *intended set of inputs* $\mathcal{X}_I := \{x \in \mathcal{X} : \mathcal{D}_I(x) > \epsilon\} \subset \text{supp}(\mathcal{D}_I)$; next we endow it with a sigma algebra $\mathcal{A}_{\mathcal{X}_I} \subset \mathcal{A}_{\mathcal{X}}$. Let $(\mathcal{Y}, \mathcal{A}_{\mathcal{Y}})$ be the measurable space of (class) labels, and assume $\mathcal{Y}$ is at most countable. The *oracle classifier* is a $\mathcal{A}_{\mathcal{X}_I} \backslash \mathcal{A}_{\mathcal{Y}}$-measurable map $C : \mathcal{X}_I \to \mathcal{Y}, x \mapsto C(x) = y \in \mathcal{Y}$, which produces the ground truth labels. With $\mathcal{X}' := \mathcal{X} \setminus \mathcal{X}_I$ we define the following space

$$\mathscr{F} := \{F : \mathcal{X} \times \mathcal{X}' \to \mathcal{X}_I, (s, z) \mapsto F(s, z) = x, C(x) = C(s)\}.$$

The elements of $\mathscr{F}$ are maps that combine semantically relevant and irrelevant images so as to preserve the original label of the image. The label is determined by the oracle classifier $C$.

Our assumptions are the following three:

1. $\mathscr{F} \neq \emptyset$. There exists at least one map $F$ that combines semantically relevant and irrelevant images so as to preserve the original label of the image.
2. $\sup_{x \in \mathcal{X}} |\mathcal{D}(x) - \mathcal{D}_I(x)| \leq \delta$, for some $\delta \geq 0$. The distance between the original and the intended distributions is bounded by a quantity $\delta$.
3. $\sup_{x \in \mathcal{X}} |\mathcal{D}(x) - \hat{\mathcal{D}}_I(x)| \leq \eta$, for some $\eta \geq 0$. The distance between $\mathcal{D}$, and an estimate of the intended distribution $\mathcal{D}_I$, that we are able to compute is bounded by a quantity $\eta$.

The first assumption is a realistic one, since it is almost always the case that we can decompose an image into the two aforementioned components. The second assumption is reasonable because it is usually the case, that the distance between the intended and the sampling distribution is not arbitrarily large. The third assumption states that when building the estimate of the intended distribution, we do not move too far away from the sampling distribution. Then, we have the following.

**Theorem 1.** *Let $\mathcal{D}_I, \hat{\mathcal{D}}_I \in \Delta(\mathcal{X}, \mathcal{A}_{\mathcal{X}})$ be defined as above. Then, there is $\nu \geq 0$ such that $d(\mathcal{D}_I, \hat{\mathcal{D}}_I) := \sup_{x \in \mathcal{X}} |\mathcal{D}_I(x) - \hat{\mathcal{D}}_I(x)| \leq \nu$.*

---

[2]As no confusion arises, we do not distinguish between probability measure and probability distribution.

---

**Algorithm 1** Detecting OOD Inputs as Out-of-Intended Distribution Inputs

---

**Input:** Test datapoint $o \in \mathcal{X}$
**Parameters:** Training set $\mathcal{S}$, detection threshold $\epsilon$
**Output:** "1" if $o$ is detected as OOD; "0" otherwise
$\hat{\mathcal{X}}_I$ = estimated $\mathcal{X}_I$ from $\mathcal{S}$
$\hat{\mathcal{D}}_I = OOD\_Detection(\hat{\mathcal{X}}_I, x \in \mathcal{X})$
Return 1 if $\hat{\mathcal{D}}_I(\hat{\mathcal{X}}_I, o) < \epsilon$, 0 otherwise

---

The proof of Theorem 1 is included in Appendix. The Theorem states that the distance between the estimated intended distribution $\hat{\mathcal{D}}_I$ and the intended distribution $\mathcal{D}_I$ is bounded by $\nu = \delta + \eta \geq 0$. If the sampling process was perfect, then intended distribution $\mathcal{D}_I$ would be arbitrarily close to $\mathcal{D}$, hence $\delta = 0$ and the distance between $\hat{\mathcal{D}}_I$ and $\mathcal{D}_I$ would be bounded by $\eta$ only. That is the error due to the short-fall of the algorithm which estimates the intended distribution from the sampled data.

**OOD Detection**: In image classification, the intended input set $\mathcal{X}_I$ is the set of images with class labels in $\mathcal{Y}$. An OOD input is an image whose (associated class) label does not belong to $\mathcal{Y}$. In light of Theorem 1, that shows that – under three natural assumptions – the distance between $\mathcal{D}_I$ and $\hat{\mathcal{D}}_I$ is bounded, we propose to perform OOD detection using $\hat{\mathcal{D}}_I$. With $\hat{\mathcal{X}}_I$ being the estimation of $\mathcal{X}_I$ obtained from training set $\mathcal{S}$, and $\hat{\mathcal{D}}_I$ being estimated from $\hat{\mathcal{X}}_I$, Algorithm 1 proposes to detect those inputs as OOD whose probability according to $\hat{\mathcal{D}}_I$ is low.

We can assign probabilities to the (class) labels associated to the elements of $\mathcal{X}_I$ using a pushforward argument. That is, we define $P_{\mathcal{Y}}$ as:

$$\mathcal{Y} \ni y \mapsto P_{\mathcal{Y}}(y) \equiv C_\sharp \mathcal{D}_I(y) := \mathcal{D}_I(C^{-1}(y)),$$

We estimate $\mathcal{D}_I$ for OOD detection. In the next section, we propose two ways of defining the OOD detection function $\mathcal{D}_I$ in Algorithm 1.

# 3 USING SEMANTICALLY RELEVANT INFORMATION FOR OOD DETECTION

In this section, we use two kinds of maps, $\mathcal{N}_s$ and $\mathcal{N}_r$. The former – introduced in section 3.1 – represents a semantic segmentation network, while the latter – introduced in section 3.2 – represents a two-step process. We estimate the set $\mathcal{X}_I$ of intended images for a classifier from the class-specific semantic information contained in $\mathcal{S}$. Inputs without semantic information for any class in $\mathcal{Y}$ are detected as OOD. We define:

$$\hat{\mathcal{X}}_I := \{x' \in \mathbb{R}^{h \times w \times 3} : x' = \mathcal{N}(x), x \in \mathcal{S}\}. \text{ where, } \mathcal{N} \text{ is a generic segmentation map.}$$

We argue that $\hat{\mathcal{X}}_I$ is a good estimator of $\mathcal{X}_I$ because it preserves the semantically relevant information that is required for classification. This is in line with our intuition of using the elements of set $\mathscr{F}$ which combine semantically relevant and irrelevant information.

## 3.1 OOD DETECTION WITH SEMANTIC SEGMENTATION NETWORK

In scenarios with large amount of labeled available data, we propose to use semantic segmentation models as $\mathcal{N}_s$. The output of a segmentation network, the segmentation map, is the classification of each pixel in the image into either background (semantically irrelevant information) or one of the class labels in $\mathcal{Y}$. We propose $\hat{\mathcal{X}}_I$ as the set of segmentation maps on (the elements of) $\mathcal{S}$, where class information is labeled by the segmentation network. We call that segment of a segmentation map as the *foreground segment* which is labeled with a class in $\mathcal{Y}$.

Here $\mathcal{N}_s$ is a segmentation algorithm that filters the input with the class-specific semantic information in $\mathcal{Y}$. It can be seen as a $\mathcal{A}_{\mathcal{X}} \backslash \mathcal{B}(\mathbb{R}^{h \times w \times (|\mathcal{Y}|+1)})$-measurable function $\mathcal{N}_s : \mathcal{X} \to \mathbb{R}^{h \times w \times (|\mathcal{Y}|+1)}$, where $\mathcal{B}(\mathbf{R})$ denotes the Borel sigma-algebra of a generic Euclidean space $\mathbf{R}$. In particular, $\mathcal{N}_s$ only keeps height and width of the image, losing the color information; the third dimension is given by a vector of dimension $|\mathcal{Y}| + 1$, where $|\mathcal{Y}|$ is the number of (class) labels, and the extra dimension captures an "extra label" associated to the background and all the images that are not assigned any of

the labels in $\mathcal{Y}$. Its entries are reals between $0$ and $1$ that sum up to $1$; they represent the probability of image $x$ belonging to (class) label $y \in \mathcal{Y}$ or to the "extra label".

**OOD Detection Scores:** The classification-based detection scores by the existing detectors (Hendrycks & Gimpel, 2016; Liang et al., 2017) can be used for OOD detection on the foreground segment of the input image. Similar to the baseline detector (Hendrycks & Gimpel, 2016), which uses softmax score of the predicted class by a classification network for detection, we propose to use softmax scores for the predicted class of the foreground segment for detection. Since the detection score has to be a single value, we take average of the softmax scores for the pixels in the foreground segment.

Recall that $\mathcal{X} \subset \mathbb{R}^{h \times w \times 3}$. Let $\mathscr{C} = \{\emptyset, \mathcal{X}, c_1, \ldots, c_N, c_{N+1}\}$, $N \in \mathbb{N}$, be a partition of $\mathcal{X}$ whose elements are $N$ classes of images (e.g. birds, clothes, cars, etc.), computed as:

$$c_j := C^{-1}(y) = \{x \in \mathcal{X}_I : C(x) = y\} \text{ for all } y \in \mathcal{Y}, j \in \{1, \ldots, N\},$$

one class – the $(N + 1)$-th – that incorporates the background and everything that is not subsumed in the other $N$ classes, that is, $c_{N+1} = \mathcal{X} \setminus (\cup_{j=1}^{N} c_j)$, and the whole space $\mathcal{X}$ and the empty set $\emptyset$.

Pick any $x \in \mathcal{X}$ and let $H = \{1, \ldots, h\}$ and $W = \{1, \ldots, w\}$. For a generic vector $a$, we denote by $a_i$ its $i$-th entry, while for a generic element $t$ of $\mathbb{R}^{h \times w \times (|\mathcal{Y}|+1)}$, we write $t_{i,j}$ to denote the $(|\mathcal{Y}|+1)$-dimensional vector that we obtain if we "slice" $t$ at first coordinate $i$ and second coordinate $j$.

Let $q = (q_1, \ldots, q_N, q_{N+1})^\top \in \mathbb{R}^{N+1}$; define function

$$q \mapsto S(q) := \begin{cases} \max_i q_i & \text{when } \arg\max_i q_i \in \{1, \ldots, N, N+1\} \\ 0 & \text{otherwise} \end{cases}.$$

**Definition 1.** *Pick any $x \in \mathcal{X}$. Define the following baseline score*

$$BS(x) := \frac{\sum_{i \in H} \sum_{j \in W} S(\mathcal{N}_s(x)_{i,j})}{h \cdot w} \tag{1}$$

We can also use the classification-based score used by the ODIN detector (Liang et al., 2017). ODIN is an enhanced version of the baseline detector where the temperature-scaled softmax score of the preprocessed input is used for detection. The input is preprocessed by adding small perturbations:

$$\widetilde{x} := x - \zeta \, \text{sign}(-\nabla_x \log S'_\star(x, T)).$$

Here $\zeta > 0$ is the perturbation magnitude, sign denotes the sign function, $T \in \mathbb{R}_{>0}$ is the temperature scaling parameter, $S'(x, T)$ is an $N$-dimensional vector whose $i$-th entry

$$S'_i(x, T) = \frac{\exp(f_i(x)/T)}{\sum_{j=1}^{N} \exp(f_j(x)/T)}$$

is given by the temperature-scaled softmax score of the $i$-th class predicted by the classification network $\mathbf{f} = (f_1, \ldots, f_N,)$ that is trained to classify classes in $\{c_1, \ldots, c_N\}$, and $S'_\star(x, T) = \max_i S'_i(x, T)$.

**Definition 2.** *Pick any $x \in \mathcal{X}$. Define the ODIN score as*

$$OS(x) := \frac{\sum_{i \in H} \sum_{j \in W} S(\mathcal{N}_s(\widetilde{x})_{i,j})}{h \cdot w}. \tag{2}$$

Algorithm 2 is the proposed algorithm for OOD detection with $\hat{\mathcal{X}}_I$ as the segmentation maps of $S$.

**Algorithm 2** $\hat{\mathcal{D}}_I$ : OOD Detection with Semantic Segmentation Network

---

**Input:** Test input $o \in \mathcal{X}$,
**Parameters:** Semantic segmentation network $\mathcal{N}_s$ trained on $\mathcal{S}$, detection score $ds \in \{BS, OS\}$, detection threshold $\epsilon$
**Output:** "1" if $o$ is detected as OOD; "0" otherwise
$o' = \mathcal{N}_s(o)$
Create $z$ s.t, $z_{i,j} = S(o'_{i,j}), \forall i \in h, \forall j \in w$
Return 1 if $ds(z) < \epsilon$, 0 otherwise

---

**Algorithm 3** $\hat{\mathcal{D}}_I$ : OOD Detection with Reference Set

---

**Input:** Test input $o \in \mathcal{X}$
**Parameters:** Segmentation algorithm $\mathcal{N}_r$, reference set $\mathcal{R}$, detection threshold $\epsilon$
**Output:** "1" if $o$ is detected as OOD; "0" otherwise
$v = ComputeNearestSample(\mathcal{R}, \mathcal{N}_r(o))$
Return 1 if $\max(v) < \epsilon$, 0 otherwise

---

## 3.2 OOD Detection with a Reference Set

Datasets such as MNIST, with a history of feature engineering techniques (Belongie et al., 2000), permit semantically relevant pixels to be derived easily. The generation of the segmentation $\mathcal{A}_{\mathcal{X}} \backslash \mathcal{B}(\mathbb{R}^{h \times w \times 3})$-measurable map $\mathcal{N}_r : \mathcal{X} \to \mathbb{R}^{h \times w \times 3}$ follows a two step process. First it uses a standard segmentation algorithm to define super-pixels of an image. Next, it removes the segments which are can be regarded as irrelevant information. We create an image out of the two components by setting different colors to these pieces. We leave the details with examples (Fig. 8) to the Appendix.

**OOD Detection Score**: Given a pair of segmented images, we compare them using a metric known as the SSIM value. It is a well known metric to compare similarity between images. OOD detection is performed by measuring its similarity with the nearest neighbor in a reference set.

**SSIM :** Structural similarity index metric (SSIM) (Wang et al., 2004) computes statistical similarity between two images. SSIM value is calculated as:

$$SSIM(x_1, x_2) = S_1(x_1, x_2)S_2(x_1, x_2), \text{ where}$$
$$S_1(x_1, x_2) = lum(x_1, x_2), S_2(x_1, x_2) = con(x_1, x_2)corr(x_1, x_2) \tag{3}$$

The functions $lum$, $con$ and $corr$ compare the luminosity, contrast and correlation between two image inputs $x_1$ and $x_2$. The details of the implementation can be found in (Wang et al., 2004) and (Brunet et al., 2012). SSIM permits fast GPU based implementation. Next, we define a reference set $\mathcal{R} \subset \hat{\mathcal{X}}_I$ as a set of size $|\mathcal{Y}|$ containing one representative of each class $y \in \mathcal{Y}$; by representative, we mean an image whose associated (class) label is some $y \in \mathcal{Y}$. It is defined as:

$$\mathcal{R} := \{x \in \hat{\mathcal{X}}_I : x \sim \text{Unif}(C^{-1}(y)), \forall y \in \mathcal{Y}\} \subset \mathcal{X}_I.$$

A more involved algorithm can be used to replace this simple choice, such as the ones proposed in Yang et al.; Dutta et al. (2022). But, we find this simple procedure to work well in this context.

**Algorithm**: Algorithm 3 combines these pieces together. We compute SSIM of an input image $o \in \mathcal{X}$ with all images in $\mathcal{R}$ and use the maximum value for detection. In other words if the similarity value of $o$ with its nearest neighbor in $\mathcal{R}$ is below the threshold $\epsilon$, we declare $o$ as OOD.

## 4 Experiments

We perform experiments with the existing state-of-the-art (SOTA) detectors from all the three categories of supervised, unsupervised, and self-supervised OOD detection techniques.

**Unsupervised :** Baseline detector (Hendrycks & Gimpel, 2016) is the SOTA unsupervised detector. It uses softmax score of a classifier for the predicted class. ODIN (Liang et al., 2017) is an enhanced version of the baseline detector that uses temperature-scaled softmax score but, for a perturbed input for detection. Details are in section 3.1.

**Supervised :** Mahalanobis detector (Mahala) is the SOTA supervised detector which uses Mahalanobis distance (Mahalanobis, 1936) of the input in the training feature space of the classifier for detection.

Figure 2: Examples of images from the three test cases for COCO.

**Self-supervised :** Aux (Hendrycks et al., 2019) is the SOTA self-supervised detector which uses error in the prediction of the applied transformation on the input for detection. It trains a classifier with an auxiliary task of predicting the an applied rotation, vertical and horizontal translations. The sum of the error in the three predictions and classification error is used for detection.

**Evaluation Metrics:** We call in-distribution inputs as positives and OOD inputs as negatives. We report the Receiver Operating Characteristic curve (ROC), Area under ROC (AUROC), and True Negative Rate (TNR) at $95\%$ True Positive Rate (TPR) for evaluation. These are the standard metrics used in OOD detection (Hendrycks et al., 2019; Liang et al., 2017; Lee et al., 2018).

### 4.1 CASE STUDY I: OOD DETECTION WITH SEMANTIC SEGMENTATION NETWORK

#### 4.1.1 DATASET AND MOTIVATION

Common Objects in Context-Stuff (COCO) dataset (Caesar et al., 2018b) is a large-scale vision dataset created for the purpose of training machine learning models for object detection, segmentation and captioning with $182$ object classes. We use that subset (training and test) of COCO which can be classified with the class labels from the set $\mathcal{Y} = \{$cup, umbrella, orange, toaster, broccoli, banana, vase, zebra, kite$\}$. These classes are common to another dataset Vizwiz (Chiu et al., 2020). Vizwiz is a real patient dataset captured by blind people. This dataset is collected with the purpose of assisting partially blind people. Where the quality of images captured can be an issue. So, images in the Vizwiz are labeled with either "no issues", or with issues such as "blurry", "too bright", "too dark", "camera obstructed" etc. We call the images with "no issues" label in the Vizwiz dataset as the *clear Vizwiz*.

We train the ResNet18 (He et al., 2016) model to classify the training set of COCO dataset. With $74.33\%$ as model's accuracy on test COCO, it achieves a comparable accuracy of $68.14\%$ on the clear Vizwiz. Detecting inputs from clear Vizwiz as OOD by the existing detectors restricts the generalizability of classifiers from the training distribution $\mathcal{D}$ to the intended distribution $\mathcal{D}_I$.

#### 4.1.2 SEMANTIC SEGMENTATION NETWORK $\mathcal{N}_s$ FOR ALGORITHM 2 AND CLASSIFIER FOR DETECTION BY EXISTING DETECTORS

As recommended by the authors of the COCO dataset (Caesar et al., 2018a), we train the DeepLab version 2 (v2) segmentation model (Chen et al., 2017) on the training set of COCO. DeepLab v2 uses ResNet101 (He et al., 2016) model as the backbone model. For a fair comparison with the existing detectors, we train the ResNet101 classifier on the training set of COCO. We use the trained classifier for OOD detection by the existing SOTA unsupervised and supervised detectors. The accuracy of the classifier on the test COCO set is $68.64\%$. COCO dataset is commonly used for object detection and segmentation. The classification accuracy of $68.64\%$ is comparable with the SOTA detection accuracy (in terms of mean average precision) of $64.2\%$ on COCO (Wei et al., 2022). For the self-supervised detector AUX, we train the ResNet101 classifier with the auxiliary losses of rotations and translations. Its classification accuracy on the test COCO set is $74.11\%$.

#### 4.1.3 TEST CASES AND RESULTS

We conduct our experiments with the following three test cases:
**(a) In-Distribution from Clear Vizwiz:** Inputs with the class labels in $\mathcal{Y}$ but from the clear Vizwiz.
**(b) OOD from Vizwiz:** Inputs with blurry, too bright, too dark, and obstructed issues from Vizwiz. Due to the quality issues of these images, this dataset cannot be labeled with any labels in $\mathcal{Y}$.
**(c) OOD From COCO:** Inputs from the test COCO dataset with class labels not in $\mathcal{Y}$. Here, we filter that subset of the test COCO that can be classified with the class labels from the set $\{$traffic light, stop sign, parking meter, fire hydrant$\}$.

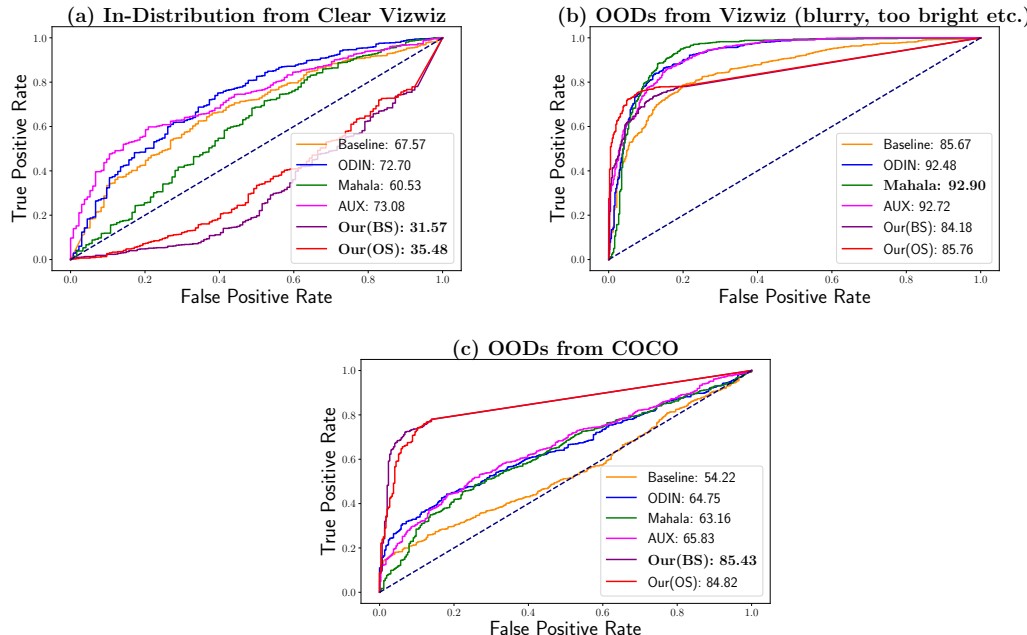

Figure 3: (a) AUROC less than 50% on in-distribution inputs from clear Vizwiz, and (c) highest AUROC on OOD inputs from COCO by Algorithm 2 shows that the proposed detection not only significantly reduces the false alarms but also improves on OOD detection on inputs with spurious features from training set.

Figure 2 shows some examples of the images from the three test cases. Figure 3 compares the ROC and AUROC results of the existing detectors with Algorithm 2 on these test cases:

**(a) In-Distribution from Clear Vizwiz (Fig. 3(a))**: AUROC less than 50% by our approach (with both the baseline and ODIN scores) implies that the proposed detector is not able to distinguish between the test COCO and Clear Vizwiz datasets. AUROC greater than 50% by the existing detectors implies that these detectors distinguish clear Vizwiz from the test COCO by assigning higher OOD detection scores to clear Vizwiz.

**(b) OOD from Vizwiz (Fig. 3(b))**: With these images as OOD for COCO, we require the AUROC to be as close to one as possible. The existing supervised detector Mahala achieves the best AUROC of 92.90% and our results are 84.18% and 85.76% with baseline and ODIN scores respectively.

**(c) OOD from COCO (Fig. 3(c))**: Significantly higher ($\geq$ 20%) AUROC by Algorithm 2 (with both scores) than the existing ones indicates that the proposed detector performs OOD detection on these inputs with spurious features from the training data better than the existing ones.

**Discussion** The performance of the proposed Algorithm 2 depends on the semantic (or foreground) information segmented in $\mathcal{X}_I$ by the segmentation network. We observe that the trained segmentation network labels all pixels in 22% images of the test COCO as background. This results in low AUROC (in comparison to 92.90% by Mahala) of 85.76% on low quality OOD inputs from Vizwiz.

If the segmentation network is trained well to segment inputs drawn from $\mathcal{X}_I$ (ex. test COCO) then the detection performance of Algorithm 2 improves. Figure 4 shows the ROC and AUROC results on the 88% images of test COCO that are segmented with class labels in $\mathcal{Y}$ by the network. This test case is analogous to using a well trained segmentation network which segments 100% of the test COCO with class labels in $\mathcal{Y}$ as improving the training of segmentation network is beyond the scope of this paper. With the 100% test COCO segmented with class labels in $\mathcal{Y}$ by the segmentation network, the performance of Algorithm 2 improves from 85.76% to 98.52% (with ODIN score) on OOD inputs from Viziwz and 85.43% to 97.42% (with baseline score) on OOD inputs from COCO. Here we show results on test cases (b) and (c). Results on test case (a) are included in Appendix.

## 4.2 CASE STUDY II: OOD DETECTION WITH A REFERENCE SET

### 4.2.1 DATASET AND MOTIVATION

We use a mixture of MNIST-M (Ganin & Lempitsky, 2015) and Background-Colored-MNIST (BC-MNIST) (Bui et al., 2021) datasets. Both MNIST-M and BC-MNIST are modified versions of

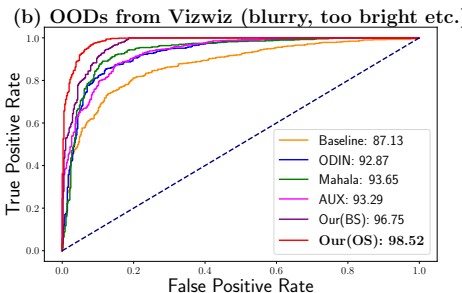 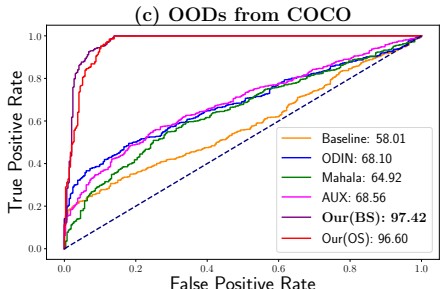

Figure 4: ROC and AUROC results of the existing detectors and Algorithm 2 on that subset of test COCO which is segmented with class labels in $\mathcal{Y}$ by the trained segmentation network $\mathcal{N}_s$.

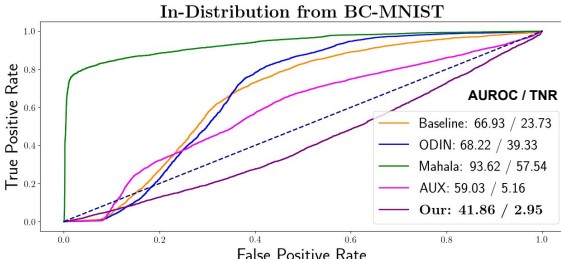

Figure 5: AUROC less than $50\%$ on in-distribution inputs from BC-MNIST by Algorithm 3 shows that the proposed algorithm significantly reduces false alarms (by the existing detectors) for training set of Mix-MNIST.

MNIST (LeCun et al., 1998) dataset. MNIST-M is MNIST with its digits blended over patches of colored images. BC-MNIST is the colored version of MNIST where both digits and background are colored. We use a mixture dataset of $100\%$ data from MNIST-M and $50\%$ data from BC-MNIST. We call this dataset as *Mix-MNIST*.

With $60,000$ training images in MNIST-M and $4000$ training images in BC-MNIST, $96.77\%$ of the training data in Mix-MNIST comes from MNIST-M and the remaining $3.23\%$ from BC-MNIST. We train the Lenet5 (LeCun et al., 1998) classifier on Mix-MNIST. The classifier achieves comparable accuracy of $90\%$ and $91\%$ on test MNIST-M and test BC-MNIST datasets respectively. Therefore, with the classifier's ability to generalize on BC-MNIST with only $3.23\%$ of BC-MNIST as the training data, detecting inputs from BC-MNIST as OOD by the exiting detectors (Fig. 5) limits the applicability of the classifier.

### 4.2.2 EXPERIMENTAL DETAILS AND RESULTS

For the existing detectors, we use trained the LeNet5 model with its accuracy of $91.91\%$ on the test set of Mix-MNIST. Figure 5 compares the ROC, AUROC, and TNR results of the existing detectors with Algorithm 3 on the test set of BC-MNIST. AUROC higher than $50\%$ by the existing detectors implies that existing detectors distinguish the test data of Mix-MNIST from the test set of BC-MNIST with higher OOD detection scores assigned to BC-MNIST. AUROC less than $50\%$ by the proposed Algorithm 3 shows that it does not distinguish between the test sets of Mix-MNIST and BC-MNIST. We achieve the lowest false alarm rate of $2.95\%$ here.

We perform additional experiments for Mix-MNIST with OOD datasets from (low quality) Vizwiz and Fashion-MNIST. Details and results on these experiments are included in Appendix.

## 5 CONCLUSION

In this paper we show that including more nuanced semantic information about the content of images can improve detection of out-of-distribution inputs. This to the best of our knowledge, is one of the first approaches which differentiates between sampling distribution and intended distribution.

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

## A    APPENDIX

### A.1    PROOF OF THEOREM 1

*Proof.* It is immediate to see that the proposed $d$ is a proper metric. Then, the proof goes as follows

$$
\begin{aligned}
d(\mathcal{D}_I, \hat{\mathcal{D}}_I) := \sup_{x \in \mathcal{X}} |\mathcal{D}_I(x) - \hat{\mathcal{D}}_I(x)| &= \sup_{x \in \mathcal{X}} |\mathcal{D}_I(x) - \hat{\mathcal{D}}_I(x) - \mathcal{D}(x) + \mathcal{D}(x)| \\
&\leq \sup_{x \in \mathcal{X}} \left\{ |\mathcal{D}(x) - \mathcal{D}_I(x)| + |\hat{\mathcal{D}}_I(x) - \mathcal{D}(x)| \right\} \\
&\leq \sup_{x \in \mathcal{X}} |\mathcal{D}(x) - \mathcal{D}_I(x)| + \sup_{x \in \mathcal{X}} |\hat{\mathcal{D}}_I(x) - \mathcal{D}(x)| \\
&= \delta + \eta.
\end{aligned}
$$

So putting $\nu = \delta + \eta$, we conclude the proof.  □

### A.2    EXAMPLE IMAGES FROM COCO SEGMENTED WITH CLASS-SPECIFIC RELEVANT INFORMATION

Figure 6 shows some examples of images sampled from COCO and clear Vizwiz datasets on the left and corresponding output of the trained semantic segmentation network $\mathcal{N}_s$ from section 4.1.2 on the right.

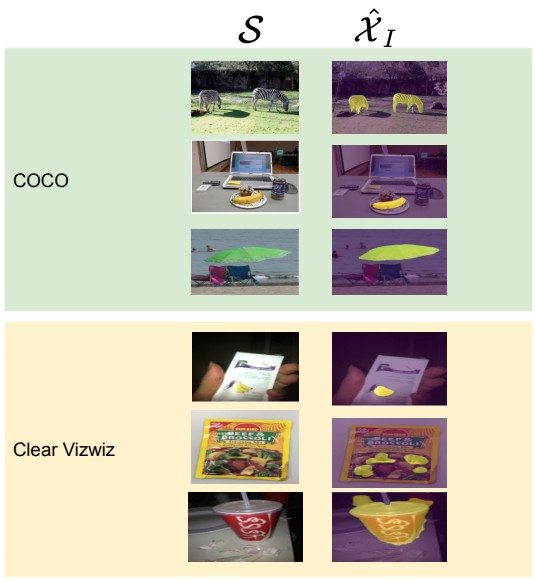

Figure 6: Example images from COCO and clear Vizwiz on left and output of the trained semantic segmentation network on these images on the right. Yellow color in the segmented images represent the class-specific semantically relevant information, and purple color represents the semantically irrelevant information.

### A.3    RESULTS ON IN-DISTRIBUTION INPUTS FROM CLEAR VIZWIZ WITH THAT SUBSET OF COCO WHICH IS SEGMENTED WITH CLASS LABELS IN $\mathcal{Y}$ BY THE SEGMENTATION NETWORK $\mathcal{N}_s$

Figure 7 shows these results. Again, AUROC less than $50\%$ by our approach (with both the baseline and ODIN scores) implies that the proposed detector is not able to distinguish between the test COCO and Clear Vizwiz datasets. AUROC greater than $50\%$ by the existing detectors implies that these detectors distinguish clear Vizwiz from the test COCO by assigning higher OOD detection scores to clear Vizwiz.

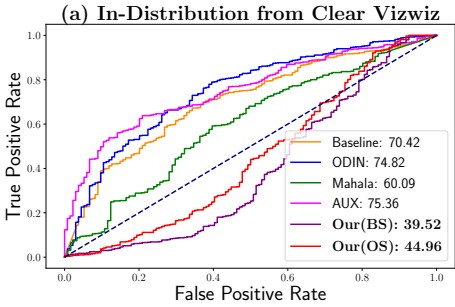

Figure 7: ROC and AUROC results of the existing detectors and Algorithm 2 on that subset of test COCO which is segmented with class labels in $\mathcal{Y}$ by the trained segmentation network $\mathcal{N}_s$.

## A.4 DETAILS ABOUT THE TWO-STEP SEGMENTATION ALGORITHM $\mathcal{N}_r$ USED IN ALGORITHM 3

Detecting semantically relevant pixels is the first step in this algorithm. In order to separate the semantically relevant pixels, we first partition the image into meaningful segments using Felzenszwalb's Algorithm Felzenszwalb & Huttenlocher (2004). Next we mark the segments placed away from the center as being semantically irrelevant. Whatever remains closely maps to semantically relevant information. We binarize the result in the previous step, to obtain a black and white version of the image. Figure 8 shows some examples of images sampled from Mix-MNIST on the left and corresponding output of the segmentation algorithm $\mathcal{N}_r$ from section 4.2.2 on the right.

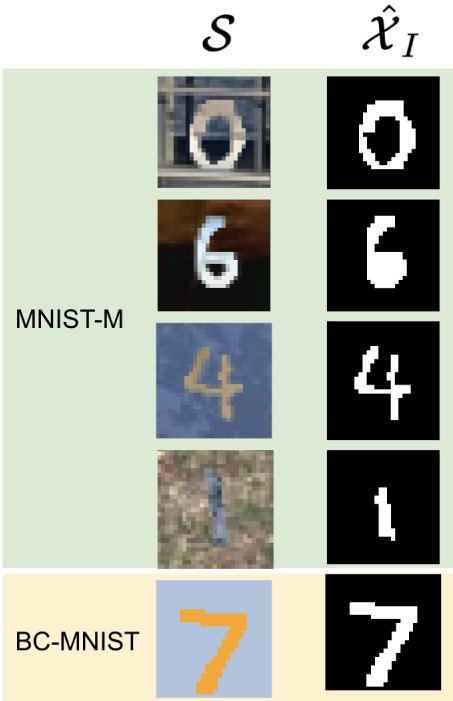

Figure 8: Example images from Mix-MNIST on left and the output of the segmentation algorithm on these images on the right. White color in the segmented images represent the class-specific semantically relevant information, and black color represents the semantically irrelevant information.

### A.5 EXPERIMENTAL DETAILS AND ADDITIONAL EXPERIMENTS ON MIX-MNIST

#### A.5.1 EXPERIMENTAL DETAILS

Given two binarized versions of an image pair by the segmentation algorithm $\mathcal{N}_r$ described in Appendix A.4, we compute the SSIM value between these images. We restrict ourselves to a non-negative version of the SSIM metric in this paper. To estimate whether an image contains digit, we maintain a reference set for digits zero to nine. Figure 9 shows the reference set used in experiments. For a given test image, we compute the SSIM between the binary version of the image and each digit image in the reference set. If the test image does not resemble any digit in the reference set, we declare it to be OOD.

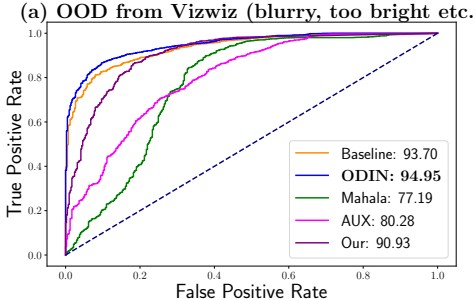

Figure 9: Reference set of 0-9 digits extracted from images in Mix-MNIST by the segmentation algorithm $\mathcal{N}_r$.

#### A.5.2 ADDITIONAL EXPERIMENTS

We conduct additional experiments on Mix-MNIST with the following two test cases:
**(a) OOD from Vizwiz:** Images with the blurry, too dark, and obstructed quality issues from Vizwiz.
**(b) OOD from Fashion-MNIST:** Images from Fashion-MNIST (Xiao et al., 2017) dataset with class labels from fashion objects such as trousers, shoe etc.

The results are as follows:
Figure 10 compares the ROC and AUROC results of the existing detectors with the proposed OOD detection Algorithm 3. Table 2 shows these results on TNR (at 95% TPR) on these test cases:
**(a) OOD from Vizwiz (Fig. 10(a))**: With failure to assign any labels to this dataset due to quality issues, these images are OOD for the Mix-MNIST dataset and here we require the AUROC to be as close to one as possible. The existing detector ODIN achieves the best AUROC of $94.95\%$ and our result is $90.93\%$. We achieve the best TNR (@95%TPR) detection of $67.15\%$ here.
**(b) OOD from Fashion-MNIST (Fig. 10(b))**: With the class labels of Fashion-MNIST disjoint from the classes in Mix-MNIST, images from Fashion-MNIST are OOD for Mix-MNIST. The existing supervised detector Mahala achieves the best AUROC of $86.03\%$ and our (unsupervised) results are comparable at $84.27\%$. Mahala achieves the best TNR (@95%TPR) detection of $56.86\%$ and ours is second best at $44.67\%$.

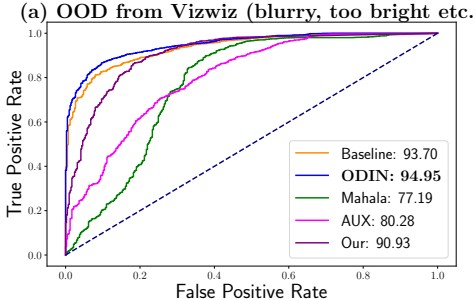
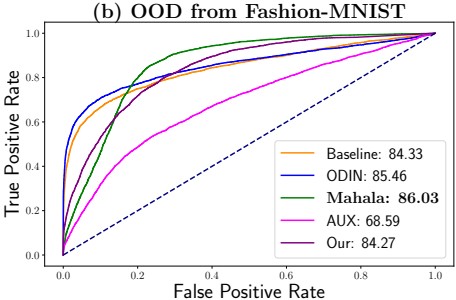

Figure 10: AUROC results of the existing detectors and Algorithm 3 on Mix-MNIST.

Table 2: TNR (@95% TPR) of existing detectors and Algorithm 3 on Mix-MNIST.

| Test Set | Baseline | ODIN | Mahala | AUX | Our |
|---|---|---|---|---|---|
| OOD from Vizwiz | 64.73 | 66.67 | 54.16 | 43.23 | **67.15** |
| OOD from Fashion-MNIST | 23.13 | 19.02 | **56.86** | 11.24 | 44.67 |

## A.6 DETAILS ON THE CELEBA AND BIRDS EXPERIMENTS

### A.6.1 CELEBA

We use the semantic segmentation network $\mathcal{N}_s$ by Lee et al. (2020) on CelebA dataset. The network segments faces into different parts including nose, hair, mouth, etc. In this experiment, we ran Algorithm 2 with the Baseline score.

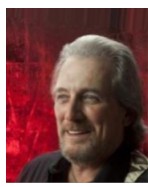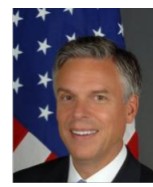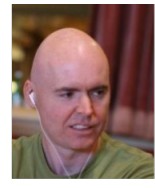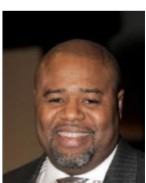

**In-distribution images:**
**Male with grey hair**

**Spurious out-of-distribution images:**
**Bald Male**

Figure 11: Images from CelebA dataset.

### A.6.2 BIRDS

We use the semantic segmentation network $\mathcal{N}_s$ by Iakubovskii (2019). Here, we select Feature Pyramid Network (FPN) (Lin et al., 2017) with ResNet50 (He et al., 2016) as its backbone architecture. It segments the images into two parts: bird and background. In this experiment, we ran Algorithm 2 with the Baseline score.

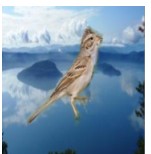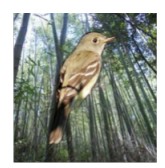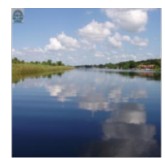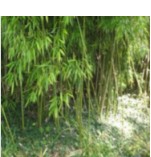

**In-distribution images:**
**waterbirds/landbirds**

**Spurious out-of-distribution images:**
**water/land images without birds**

Figure 12: Images from Birds dataset

