# OpenReview forum: "Current Anomaly Detectors are Anomalous: On Semantic Treatment of OOD Inputs"
_ICLR.cc/2023/Conference — Submitted to ICLR 2023_

### Official Review · Reviewer_sQTF · 2022-10-24

**Confidence:** 3
**Correctness:** 2
**Technical Novelty And Significance:** 2
**Empirical Novelty And Significance:** 3
**Recommendation:** 5

**Clarity, Quality, Novelty And Reproducibility:**

### Clarity

This paper is written in a clear language but was hard for me to follow and I needed a few passes.
Since this might be the due to the way the mathematical expressions are not fluent in the text, e.g. the one I mention above:
the definition of S' in the end of definition 1: stating it is a softmax *before* the expression might make the sentence more fluent.

---

### Quality
This paper seems to ignore some recent works that are not post-hoc.
For example, [Energy-based Out-of-distribution Detection](https://proceedings.neurips.cc/paper/2020/hash/f5496252609c43eb8a3d147ab9b9c006-Abstract.html) which shows that the softmax score is not the best predictor of OOD - is this the column in Table 1? if so, please add the citation.

---

### Novelty

This paper tackles a known problem but defines it in a formal manner with the assumptions leading to Theorem 1.
The proposed baseline approaches are made of known existing building blocks - so he main novelty here, hence, is the combination thereof to solve this problem.

---

### Reproducibility

This paper uses off-the-self tools to construct the baselines. Reproducing the described methods and experiments will not be easy but certainly possible for a talented grad student.

**Strength And Weaknesses:**

### Strength
This paper shows a failure case for existing approaches on semantially similar data, and that a seemingly naïve approach is much less susceptible to such a case.

The paper attempts to formalize the problem in a very rigorous manner, based on 3 reasonable assumptions in the bottom of page 3.


### Weaknesses
It is not clear to me that the proposed approaches actually estimate the intended distribution.
While theorem 1 is very attractive, I am missing
- A design choice that shows the proposed algorithms are alinged with the 3 assumptions, or in any way aimed at the elusive distribution?
 - (in light of theorem 1) why is estimating the training distribution dissimilar than estimating the intended one?
- And therefore - a non-empirical indication that prior works do not actually estimate the intended
- For the reference set baseline - why is selection of **one** representative enough? is there a theoretical justification?

Smaller notes
- clarity of notation, e.g. try to avoid using x' as the result of the map N(x) - this is confusing
- the definition of S' in the end of definition 1 is hard to read - stating it is a softmax *before* the expression might make the sentence more fluent.
- "Unif" in the end of the SSIM section is not defined.
- page 5 line 2 - did you mean "of _pixel_" rather than "of _image_"?

**Summary Of The Paper:**

This paper raises a gap in current OOD detection approaches that rely too heavily on the training examples, and falsely mark images out of the train set but with similar semantic meaning.

The authors coin the term "intended distribution" to capture both the training set, but also the rest of the "semantically similar" data.
Two baseline approaches are proposed to close this gap and estimate the intended distribution:
- using semantic segmentation, or more precisely the mean confidence of the resulting map. two flavours are proposed: either directly (BS) or via the ODIN method (OS) as a preprocess to the map.
- using a reference set, or more precisely find the most similar reference image in the SSIM sense and use the latter measure as an OOD indicator.

The experimental section compares the proposed methods to several SOTA methods from three categories of OOD techniques: supervised, unsupervised, and self-supervised.

The proposed baselines are

**Summary Of The Review:**

The paper shows a failure case of exiting OOD approaches and naive baselines that compete well in these scenarios.
While I possibly misunderstand some parts of the theoretical claims (as in- can the intended actually be estimated), the empirical part should be enough to create a discussion in the community.

---

> ### Author Response · Authors · 2022-11-18
> **Reply to Reviewer sQTF**
>
> Q - It is not clear to me that the proposed approaches actually estimate the intended distribution. While theorem 1 is very attractive, I am missing
>
> 1. A design choice that shows the proposed algorithms are aligned with the 3 assumptions, or in any way aimed at the elusive distribution?
>
> Ans - Algorithm 1 is based on the assumption that we can estimate the intended (or in) distribution from the training distribution. This requires that the intended distribution is not too far away from the training distribution (assumption 2). Algorithms 2 and 3 use the estimated in-distribution for OOD detection. This requires a bounded distance between the intended and the estimated intended distribution (Theorem 1). Theorem 1 requires both:
>
> Assumption 2:  The intended distribution is not too far away from the training distribution. This is true because, in practice it is the case that there are some samples which belong to both the intended-distribution and  the training dataset.
>
> Assumption 3:  The estimated intended distribution is not too far away from the training distribution.
>
> Assumption 1 makes sure that the images sampled from in-distribution are relevant, i.e. the original label of image is not lost due to semantically irrelevant information in these images. For ex., the input in-distribution image is not so blurry or obstructed that we cannot tell it's label.
>
> Q - (in light of theorem 1) why is estimating the training distribution dissimilar than estimating the intended one?
>
> Ans - If the training dataset is rich or sufficiently large enough to capture the (intended) in-distribution, then the training distribution represents the in-distribution. But practically, with the limited amount of training data sampled from the in-distribution, this might not always be feasible. For ex., with the training dataset of birds sitting on trees for a classifier for classifying birds in {sitting birds, flying birds}, the intended distribution for sitting birds is the one with birds sitting on trees, snow, water etc. Here the intended distribution is the distribution of images containing sitting birds irrespective of the place where the bird is sitting, whereas the training distribution is the distribution of images containing birds sitting on trees.
>
> Q - And therefore - a non-empirical indication that prior works do not actually estimate the intended
>
> Ans - Prior works estimate in-distribution by overfitting to the training data, and as shown in experiments on COCO dataset, these (existing) detectors label inputs lying outside the estimated training distribution (estimated with the bias in data collection process) but with the same training labels (lying inside the intended distribution) from another dataset such as Vizwiz as OOD.
>
> Q - For the reference set baseline - why is selection of one representative enough? Is there a theoretical justification?
>
> Ans - We can use more than one representative. For ex. 1 and  | can be used to represent the digit 'one'. In this case, an input can be labeled as OOD if it’s distance from all the representatives is higher than the detection threshold.
>
> Q - page 5 line 2 - did you mean "of pixel" rather than "of image"?
>
> Ans -  No, we mean “of image”: $\cal{X}$ is the set of images, and $x$ belongs to $\cal{X}$.
>
> Q - This paper seems to ignore some recent works that are not post-hoc. For example, Energy-based Out-of-distribution Detection which shows that the softmax score is not the best predictor of OOD - is this the column in Table 1? if so, please add the citation.
>
> Ans - Yes, Energy-based OOD detector is used in Table 1. We will cite all the detectors used  in Table 1.

---

> > ### Comment · Reviewer_sQTF · 2022-12-06
> > **Still leaning towards not accepting**
> >
> > I thank the authors for taking the time to write the detailed reply.
> >
> > The reply do not seem to answer my concerns, perhaps I was misunderstood in my questions.
> > Specifically, while the notion of the intended distribution seems clear, it is not explicitly indicated what part of the proposed algorithms stem from theorem 1 and gives the ability to estimate  intended distribution. Moreover, it is not explicitly indicated how prior methods differ from the proposed ones in their **inability** to estimate it.

---

### Official Review · Reviewer_iQtU · 2022-10-25

**Confidence:** 4
**Clarity, Quality, Novelty And Reproducibility:** Please see the above Strength And Wea…
**Correctness:** 3
**Technical Novelty And Significance:** 2
**Empirical Novelty And Significance:** Not applicable
**Recommendation:** 3

**Strength And Weaknesses:**

The proposed problem is meaningful and interesting. However, here are some questions:
- This paper attempted to introduce the complex notations in Section 2 to define semantic information for OOD detection. However, it is unclear what the exact definition of semantics is. \delta and \eta in assumptions are not sufficient to clarify the meaning of semantics. It would be good to share detailed examples for the given definition.
- In addition, the notations in Section 2 and 3 would be better to be simplified and the writing of this part can be significantly improved to make it more understandable and readable.
- In OOD detection, the proposed method is heavily dependent on the segmentation network. The experiments are tested on relatively simple datasets and show improvements. In practice, what if the dataset is complex and the segmentation network does not perform well? Moreover, compared with OOD detection, segmentation does not seem to be simpler to solve than OOD detection when data has a large variety. It is not very convincing to heavily rely on segmentation results to detect OOD.
- The experiments should be conducted on more complex datasets. The evaluations on the same datasets in existing OOD papers are expected.

**Summary Of The Paper:**

This paper argues that semantic information should be considered for OOD detection which should not be just tied to the training data distribution. This argument is interesting and practically useful. Then the authors proposed to leverage the semantic segmentation network and reference set to detect OOD data semantically. Experiments are done on COCO and MNIST datasets.

**Summary Of The Review:**

Please see the above Strength And Weaknesses section.

---

> ### Author Response · Authors · 2022-11-18
> **Reply to reviewer iQtU**
>
> Q - The experiments should be conducted on more complex datasets. The evaluations on the same datasets in existing OOD papers are expected.
>
> Ans - We ran experiments with the following OOD datasets from the prior OOD detection methods - SVHN, Imagenet and LSUN. The following table compares our AUROC results (Algorithm 2 with baseline score) on 100% test COCO (in-distribution dataset) with the baselines:
>
> -------------------------------------------------------------------------------------------------------------------------------------------------
> OOD        $\ \ \ \ \ \ \ \ \ \ \ \ \ $    |           Ours  (Baseline Score) |  Baseline  | ODIN  | Mahala (Sup)
>
> --------------------------------------------------------------------------------------------------------------------------------------------------
>
> SVHN  $\ \ \ \ \  \ \ \ \ \ \  \ $  |    $\ \ \ \ \  \ \ \ \ \ \  \ $             83.93 $\ \ \ \ \  \ \ \ \ \ \ $     |$ \ \ \ $ 87.68 $ \ \$  | 94.18 | 99.18
>
> --------------------------------------------------------------------------------------------------------------------------------------------------
>
>
> Imagenet $ \ \ \ \ \  \ $  |  $\ \ \ \ \  \ \ \ \ \ \  \ $   84.91   $\ \ \ \ \  \ \ \ \ \ \ $    | $ \ \ \$   84.51 $ \ \$  |  91.23   | 95.82
>
> --------------------------------------------------------------------------------------------------------------------------------------------------
>
>
> LSUN $\ \ \ \ \  \ \ \ \ \ \  \ $ |   $\ \ \ \ \  \ \ \ \ \ \  \ $  88.23  $\ \ \ \ \  \ \ \ \ \ \ $      | $ \ \ \$  81.18 $ \ \$   |  90.80  | 96.16
>
> --------------------------------------------------------------------------------------------------------------------------------------------------
>
> The following table compares the AUROC results on that subset of test COCO that was segmented with in-distribution classes by the segmentation network:
>
> -------------------------------------------------------------------------------------------------------------------------------------------------
> OOD        $\ \ \ \ \ \ \ \ \ \ \ \ \ $    |           Ours  (Baseline Score) |  Baseline  | ODIN  | Mahala (Sup)
>
> --------------------------------------------------------------------------------------------------------------------------------------------------
>
> SVHN  $\ \ \ \ \  \ \ \ \ \ \  \ $  |    $\ \ \ \ \  \ \ \ \ \ \  \ $            97.25 $\ \ \ \ \  \ \ \ \ \ \ $     |$ \ \ \ $ 89.08 $ \ \$  | 94.23 | 99.04
>
> --------------------------------------------------------------------------------------------------------------------------------------------------
>
>
> Imagenet $ \ \ \ \ \  \ $  |  $\ \ \ \ \  \ \ \ \ \ \  \ $  97.31  $\ \ \ \ \  \ \ \ \ \ \ $    | $ \ \ \$   86.02 $ \ \$  |  91.59   | 95.68
>
> --------------------------------------------------------------------------------------------------------------------------------------------------
>
>
> LSUN $\ \ \ \ \  \ \ \ \ \ \  \ $ |   $\ \ \ \ \  \ \ \ \ \ \  \ $  99.57 $\ \ \ \ \  \ \ \ \ \ \ $      | $ \ \ \$  83.24 $ \ \$   |  91.24  | 94.28
>
> --------------------------------------------------------------------------------------------------------------------------------------------------
>
> These results are consistent with the results reported in the paper with OOD images from Vizwiz dataset - blurry, too bright, too dark, obstructed images.

---

> > ### Author Response · Authors · 2022-11-18
> > **Reply to reviewer iQtU (Continued..)**
> >
> > Q - This paper attempted to introduce the complex notations in Section 2 to define semantic information for OOD detection. However, it is unclear what the exact definition of semantics is. \delta and \eta in assumptions are not sufficient to clarify the meaning of semantics. It would be good to share detailed examples for the given definition.
> >
> > Ans - In the paper (Section 2.1) we refer to the  semantically relevant and irrelevant pixels, and that an image is some combination of the two. Our solution to capture the intended distribution is to separate these 2 components from an image and focus on the relevant part for OOD detection. We use a semantic segmentation network to create this separation.  The notation is introduced to convey the idea that intended-distribution is usually distinct from the training distribution. But their relative distances are finite.  $\eta$ and $\delta$ are the bounds that we assume, and do not allude to the semantic information. Our hypothesis is that the gap between intended and training distribution  is created due to lack of semantic information. This, as we observe in the experiments, seems to bridge the gap between intended and actual distribution to some extent.
> >
> > Q - The experiments are tested on relatively simple datasets and show improvements. In practice, what if the dataset is complex and the segmentation network does not perform well? Moreover, compared with OOD detection, segmentation does not seem to be simpler to solve than OOD detection when data has a large variety. It is not very convincing to heavily rely on segmentation results to detect OOD.
> >
> > Ans - We discuss that the performance of Algorithm 2 depends on the semantic information segmented by the segmentation network  under “Discussion” in section 4.1.3 on page 8. We propose using semantically relevant information for OOD detection. Using segmentation networks is one way to extract this information for detection (Algorithm 2).  With the expert knowledge available for defining the semantically relevant information, we can use this knowledge for extracting the relevant information required for detection (Algorithm 3).

---

### Official Review · Reviewer_mNjq · 2022-10-27

**Confidence:** 4
**Correctness:** 1
**Technical Novelty And Significance:** 2
**Empirical Novelty And Significance:** 2
**Recommendation:** 3

**Clarity, Quality, Novelty And Reproducibility:**

The writing is pretty clear. But the paper falls short on empirical results. Much more experimentation is needed and they failed to convincingly show that their OOD detector is superior to existing ones.

They did show that existing OOD detectors can report some data as OOD even if the classifier can classify it with reasonable accuracy. But, they failed to acknowledge that  their OOD detector can conversely treat data as non-OOD but the model cannot classify it properly.

**Strength And Weaknesses:**

The thesis in the paper is provocative in that it is arguing that all the previous OOD detectors are based on a faulty premise that the training data reflects the true distribution (unknown) and thus all OOD detectors are biased; furthermore, they argue that the bias can be corrected -- at least for image classifiers.

I think that any bias in the training data is going to be reflected in the OOD detector is a reasonable thing to expect -- so that statement is not a surprise. That is true for machine learning models trained on a dataset as well -- the models have a similar bias. That is why having good training sets is usually important.

But, the paper also proposes algorithms which appear to do a better OOD detection. Unfortunately, this is where the paper falls seriously short.   The authors essentially make a one-sided argument that their OOD detector is a better detector overall because it does not classify the custom dataset that the authors constructed (which has high accuracy) on a custom model that the authors trained with blended data as OOD dataset.

But, it is also likely trivial to construct counter-examples for the authors' OOD detector below:

Author's Premise: The authors used a model that was trained on  96.77% MNIST-M and 3.23%  BC-MNIST dataset and argued that it still accurately classifies BC-MNIST dataset, but several existing OOD detectors classify the BC-MNIST dataset as OOD, but their OOD detector does not.  Therefore, their OOD detector is superior.

Counter-example to the author's premise: Let's say instead the model was trained 100% on MNIST-M (or even more extreme case, 100% original MNIST). It seems that the OOD detector by the authors would still call BC-MNIST data as in-distribution because of all the pre-processing described in the appendix that does segmentation into foreground/background and converts colors to black and white.  But in that case, I suspect either of the ML models would perform horribly on the BC-MNIST dataset while the authors' model would predict the BC-MNIST as in-distribution dataset.  So, overall, the experiments are not conclusive in showing the value of the approach.

The OOD algorithm by the authors is also complex and has a lot of priors built-in. For instance, use of segmentation to separate foreground and background and then removal of colors makes assumptions about the domain. For instance, the authors assume that segments placed away from the center are semantically irrelevant (A.4). These are priors. If they were known, potentially, a classifier should be trained on pre-processed data.  In that case, existing OOD detectors should work well since pre-processing would be part of the model.

I also encourage authors  to evaluate their OOD on situations on which prior OOD methods were evaluated.



**Summary Of The Paper:**

The paper makes the argument that current OOD detectors are faulty since they are tied to the bias in the data collection process. The authors show examples where several existing OOD detectors would have classified an auxiliary test data set as OOD, but the classifier performs well on those examples.  The authors argue that the auxiliary test data set should not have been OOD in the first place since humans could have classified the data correctly using the class labels for the original dataset.

They also propose a new OOD detector that does not classify that auxiliary dataset as OOD. Some theoretical basis for the results is also presented.





**Summary Of The Review:**

Interesting idea, but validation falls short.

---

> ### Author Response · Authors · 2022-11-18
> **Reply to the reviewer mNjq**
>
> Q - I think that any bias in the training data is going to be reflected in the OOD detector is a reasonable thing to expect. That is true for machine learning models trained on a dataset as well -- the models have a similar bias. That is why having good training sets is usually important.
>
> Ans - Though we agree with the reviewer that any degree of bias in the dataset will be reflected in the models, it misses a subtle point. We do not want the model to be applicable ONLY in the training dataset. This is strictly the opposite of the kind of behavior we desire from a system. What we discover in our experiments is that even though models can be functional outside of the training dataset, the OOD detectors are not. This motivates our paper. Another notable point is that domain generalization (Zhou et al. 2022) is an active research area where efforts are made for generalizability of machine learning classifiers to its classes beyond the training data. It tries to ask the question of whether a classifier trained on the images of birds on trees would work on images of birds on water? Domain-invariant representation learning (Li et al., 2018), training data augmentation with higher variability (Zhou et al., 2020) etc. have been proposed to solve this problem. There is a clear trend in trying to make machine learning classifiers function beyond the training data.
>
> Q - The authors essentially make a one-sided argument that their OOD detector is a better detector overall because it does not classify the custom dataset that the authors constructed (which has high accuracy) on a custom model that the authors trained with blended data as OOD dataset.
>
> Ans - We would like to correct the reviewer here, Common Objects in Context-Stuff (COCO) is a well known dataset. As recommended by the authors of the COCO dataset (Caesar et al., 2018), we train the DeepLab version 2 (v2) segmentation model (Chen et al., 2017) on the training set of COCO. We did not blend the trained model with Vizwiz (Chiu et al., 2020) - another well known dataset with pictures clicked by partially blind people. It is publicly available on https://vizwiz.org/ and has a practical utility of assisting blind people with the quality of the pictures clicked by them. The segmentation model or the classification ResNet101 model (for baselines) never saw Vizwiz during their training. So, Vizwiz is:
>
> 1. not a custom dataset created by us, and
>
> 2. not blended by the any custom model trained by us
>
> We also report low OOD detection by existing SOTA detectors on OOD inputs with spurious features from the training data of Birds and CelebA datasets in the Introduction. These results in Table 1 are from Ming et al., 2022, and not customized by us . Here, we show that our detector achieves significant improvement of 57.22% and 45.64% on OOD detection for Birds and CelebA, respectively.
>
> Q - Counter-example to the author's premise: Let's say instead the model was trained 100% on MNIST-M (or even more extreme case, 100% original MNIST). It seems that the OOD detector by the authors would still call BC-MNIST data as in-distribution because of all the pre-processing described in the appendix that does segmentation into foreground/background and converts colors to black and white. But in that case, I suspect either of the ML models would perform horribly on the BC-MNIST dataset while the authors' model would predict the BC-MNIST as an in-distribution dataset. So, overall, the experiments are not conclusive in showing the value of the approach.
>
> Ans - This is precisely the point where we believe the reviewer is mistaken.  We wish to build models which can generalize and so should the OOD detectors. In cases when the models CAN generalize, even the OOD detectors should. The SOTA OOD detectors do not have the metrics or ability to stretch its ability in this fashion. We additionally observe this  experimentally as well.   As suggested by the reviewer, we trained a classifier on 100% MNIST-M dataset and tested it on BC-MNIST. The classifier’s accuracy on BC-MNIST is 67.07%. These results bolster our point even further.

---

> > ### Author Response · Authors · 2022-11-18
> > **Reply to the reviewer mNjq (Continued..)**
> >
> > Q - The OOD algorithm by the authors is also complex and has a lot of priors built-in. For instance, use of segmentation to separate foreground and background and then removal of colors makes assumptions about the domain. For instance, the authors assume that segments placed away from the center are semantically irrelevant (A.4). These are priors. If they were known, potentially, a classifier should be trained on pre-processed data. In that case, existing OOD detectors should work well since pre-processing would be part of the model.
> >
> > Ans - We think that the reviewer is potentially confused between Algorithm 2 and Algorithm 3. We propose using a segmentation network for extracting semantically relevant information representing the intended in-distribution in Algorithm 2. With the domain knowledge or priors available for defining the semantically relevant information, we propose to use these priors for defining the semantically relevant information for detection in Algorithm 3. Removal of colors and details in A.4 are on the experiments for Algorithm 3 on MNIST dataset with the known priors.  Additionally, even if it is at times possible to extract some semantic information, it is not enough for classification and achieving good accuracy. This prohibits us from making the logical jump alluded to by the reviewer : even if a pre-processing exists it is not clear that it can be included in the model. But as we demonstrate, it can be included into the OOD detection pipeline.
> >
> > Q - I also encourage authors to evaluate their OOD on situations on which prior OOD methods were evaluated.
> >
> > Ans - We ran experiments with the following OOD datasets from the prior OOD detection methods - SVHN, Imagenet and LSUN. The following table compares the AUROC results on 100% test COCO (in-distribution dataset):
> >
> > -------------------------------------------------------------------------------------------------------------------------------------------------
> > OOD        $\ \ \ \ \ \ \ \ \ \ \ \ \ $    |           Ours  (Baseline Score) |  Baseline  | ODIN  | Mahala (Supervised detector)
> >
> > --------------------------------------------------------------------------------------------------------------------------------------------------
> >
> > SVHN  $\ \ \ \ \  \ \ \ \ \ \  \ $  |    $\ \ \ \ \  \ \ \ \ \ \  \ $             83.93 $\ \ \ \ \  \ \ \ \ \ \ $     |$ \ \ \ $ 87.68 $ \ \$  | 94.18 | 99.18
> >
> > --------------------------------------------------------------------------------------------------------------------------------------------------
> >
> >
> > Imagenet $ \ \ \ \ \  \ $  |  $\ \ \ \ \  \ \ \ \ \ \  \ $   84.91   $\ \ \ \ \  \ \ \ \ \ \ $    | $ \ \ \$   84.51 $ \ \$  |  91.23   | 95.82
> >
> > --------------------------------------------------------------------------------------------------------------------------------------------------
> >
> >
> > LSUN $\ \ \ \ \  \ \ \ \ \ \  \ $ |   $\ \ \ \ \  \ \ \ \ \ \  \ $  88.23  $\ \ \ \ \  \ \ \ \ \ \ $      | $ \ \ \$  81.18 $ \ \$   |  90.80  | 96.16
> >
> > --------------------------------------------------------------------------------------------------------------------------------------------------
> >
> > The following table compares the AUROC results on that subset of test COCO that was segmented with in-distribution classes by the segmentation network:
> >
> > -------------------------------------------------------------------------------------------------------------------------------------------------
> > OOD        $\ \ \ \ \ \ \ \ \ \ \ \ \ $    |           Ours  (Baseline Score) |  Baseline  | ODIN  | Mahala (Supervised detector)
> >
> > --------------------------------------------------------------------------------------------------------------------------------------------------
> >
> > SVHN  $\ \ \ \ \  \ \ \ \ \ \  \ $  |    $\ \ \ \ \  \ \ \ \ \ \  \ $            97.25 $\ \ \ \ \  \ \ \ \ \ \ $     |$ \ \ \ $ 89.08 $ \ \$  | 94.23 | 99.04
> >
> > --------------------------------------------------------------------------------------------------------------------------------------------------
> >
> >
> > Imagenet $ \ \ \ \ \  \ $  |  $\ \ \ \ \  \ \ \ \ \ \  \ $  97.31  $\ \ \ \ \  \ \ \ \ \ \ $    | $ \ \ \$   86.02 $ \ \$  |  91.59   | 95.68
> >
> > --------------------------------------------------------------------------------------------------------------------------------------------------
> >
> >
> > LSUN $\ \ \ \ \  \ \ \ \ \ \  \ $ |   $\ \ \ \ \  \ \ \ \ \ \  \ $  99.57 $\ \ \ \ \  \ \ \ \ \ \ $      | $ \ \ \$  83.24 $ \ \$   |  91.24  | 94.28
> >
> > --------------------------------------------------------------------------------------------------------------------------------------------------
> >
> > These results are consistent with the results reported in the paper with OOD images from Vizwiz dataset - blurry, too bright, too dark, obstructed images.

---

> > > ### Author Response · Authors · 2022-11-18
> > > **Reply to reviewer mNjq (Continued)**
> > >
> > > Q - They did show that existing OOD detectors can report some data as OOD even if the classifier can classify it with reasonable accuracy. But, they failed to acknowledge that their OOD detector can conversely treat data as non-OOD but the model cannot classify it properly.
> > >
> > > Ans - We tested the percentage of the non-OOD inputs (in-distribution inputs not detected as OODs by $\textbf{our detector}$) misclassified by the Lenet classifier trained and used in the paper for classifying Mix-MNIST. Only 5.60% of the test in-distribution inputs correctly detected as in-distribution were misclassified by the classifier.
> > >
> > > We will add these results and acknowledge the concern of the reviewer in the final version of the paper. Please note that whether we acknowledge the second or the first statement, it remains the case that SOTA OOD detectors are not simply lacking in being the best OOD detectors, the general research direction of what  OOD detectors are striving for, is potentially misaligned with reality. In this paper we are realigning the research field in what we believe would be the proper set of experiments to test OOD detectors on.

---

> > ### Comment · Reviewer_mNjq · 2022-11-22
> > **dataset question**
> >
> > In the Vizviz dataset, how do I find No Issue Images that are labeled as Zebras? I searched at the dataset's web site, but I only located 7 images that had zebra in the caption. They also included images that were a  vase with a striped pattern as well as a pen. I am sure I am missing something. Are you arguing that they should not be OOD with respect to the COCO dataset, since they have the word zebra in the caption?
> >
> > The customization I was referring to was not regarding the COCO dataset, but datasets used in Section 4.2.1 (e.g., see first para there).
> >
> > Thanks for other clarifications -- I will think about them, but the answers didn't always clarify my doubts. In the Counter-example question, for instance, you didn't say clearly whether your OOD detector would call BC-MNIST data as in-distribution or not and to what extent. I do see that the classifier accuracy dropped to 67%, which seems like a significant drop.

---

> > > ### Author Response · Authors · 2022-11-22
> > > **Reply to the dataset question**
> > >
> > > Q - In the Vizviz dataset, how do I find No Issue Images that are labeled as Zebras? I searched at the dataset's web site, but I only located 7 images that had zebra in the caption. They also included images that were a vase with a striped pattern as well as a pen. I am sure I am missing something. Are you arguing that they should not be OOD with respect to the COCO dataset, since they have the word zebra in the caption?
> > >
> > > Ans - We downloaded images of interest (ex. captions containing zebra) from the Vizwiz dataset using their API: https://github.com/Yinan-Zhao/vizwiz-caption.  After downloading, we manually verified and used only that subset of images that actually contain the object of interest such as zebra. We did not use the images such as the one that the reviewer mentioned - vase with striped pattern as well as a pen. We are not arguing that images with zebra print are not OOD with respect to COCO since these images contain the word ‘zebra’ in the caption. We are arguing that for the model trained with images of zebra from COCO dataset, images from Vizwiz containing zebra are not OOD.
> > >
> > > Q -  In the Counter-example question, for instance, you didn't say clearly whether your OOD detector would call BC-MNIST data as in-distribution or not and to what extent.
> > >
> > > Ans - For a model trained on MNIST-M, we do not want images from BC-MNIST to be detected as OOD.

---

### Official Review · Reviewer_fcgs · 2022-10-31

**Confidence:** 4
**Correctness:** 3
**Technical Novelty And Significance:** 3
**Empirical Novelty And Significance:** 2
**Recommendation:** 5

**Clarity, Quality, Novelty And Reproducibility:**

+ The paper is overall well written and easy-to-follow.
+ The paper seems to include all the details necessary to reproduce the presented results.

* I find the novelty of the paper to be fair, though OOD generalization vs. detection has also been discussed in previous works.

- The paper includes some technical inaccuracies, e.g. an "AUROC less than 50% [...] implies that the proposed detector is not able to distinguish between the test COCO and Clear Vizwiz datasets." is false. A detector cannot distinguish and is random at an AUROC of 50%. Below 50%, the ranking becomes inverse.

---

*Additional Comments*
* In the definition of the function $S$, the first case should only be on $\{1, ..., N\}$, right?
* Strictly speaking, $C$ in the definition of $\mathcal{F}$ is not defined on $\mathcal{X}'$, but only $\mathcal{X}$, right?

**Strength And Weaknesses:**

*Strengths*
+ The paper contributes some helpful conceptual structure for thinking about OOD detection, essentially discussing the balance of OOD detection vs. generalization, arguing that OOD detection should generalize to semantically coherent classes (disregarding potentially spurious background correlations)
+ The paper is well motivated and integrated into the existing literature.

*Weaknesses*
- The proposed method essentially defers OOD detection for classification to a segmentation problem, which requires more granular information (pixel-/object-level annotations). I think this aspect is not communicated clearly in the paper yet, and I think to some degree the reported improvements are expected, given the proposed model uses much more semantic information from a segmentation model.
- On the other hand, the paper does not seem to put in much effort in testing the limits of well trained segmentation models ("[...] as improving the training of segmentation network is beyond the scope of this paper."), which presents the foundation of the proposed approach. Here, I think the paper should go deeper in exploring the strengths and limits of this approach.

**Summary Of The Paper:**

The paper presents a discussion on the definition of the in-distribution (ID) vs. out-of-distribution (OOD) for OOD detection problems, arguing that existing methods fail to semantically extrapolate the in-distribution (e.g. the ID of flying and sitting birds should include and generalize to various backgrounds, e.g. sky, woods, water, etc.). The paper calls such a "semantically complete" characterization of the ID the "Intended Distribution". Following this argument, the paper proposes to leverage semantic segmentation networks for OOD detection generalizing to the intended distribution. Here, classification OOD methods (e.g. maximum softmax or ODIN) are proposed to be used over the pixels in the foreground, whereas pixels in the background attain the max OOD score. An experimental evaluation on COCO and Vizwiz is presented showing that the proposed methods compares favorably over previous approaches in the intended distribution OOD detection task.

**Summary Of The Review:**

The paper presents some interesting and valid discussion on OOD detection vs. generalization, proposing to leverage segmentation networks to improve OOD detection that generalizes to semantically coherent classes. However, cost aspects of requiring more granular information for segmentation are not yet sufficiently addressed and discussed in my opinion.

---

> ### Author Response · Authors · 2022-11-18
> **Reply to Reviewer fcgs**
>
> Q - The paper includes some technical inaccuracies, e.g. an "AUROC less than 50% [...] implies that the proposed detector is not able to distinguish between the test COCO and Clear Vizwiz datasets." is false. A detector cannot distinguish and is random at an AUROC of 50%. Below 50%, the ranking becomes inverse.
>
> Ans - Training distribution is the distribution of the images in the COCO dataset. Images from Vizwiz are not in training data but are in-distribution. Our goal is to ensure that detection scores on all in-distribution samples (including Vizwiz which are not in the training data) are such that these are not detected as OOD. This is be possible when:
> 1) The scores on the in-distribution samples not in training data (Vizwiz) are indistinguishable from these scores of the samples in training dataset (COCO) - here AUROC will be 50%.
> 2) The scores on the in-distribution samples not in training data (Vizwiz)  are even better than these scores of the samples in the training dataset - here AUROC is bound to be less than 50%.
>
> Therefore AUROC below 50% is not a flaw but an artifact of our experiments. We just need to make sure that AUROC shouldn't be more than 50% in which case we will falsely classify in-distribution samples that are not in the training dataset as OOD.
>
> Q - In the definition of the function S, the first case should only be on 1,...,N, right?
>
> Ans - Yes, that is correct. [We will rectify the definition of S(q) on page 5.]
>
> Q - Strictly speaking, C in the definition of F is not defined on X′, but only X, right?
>
> Ans -  Yes, C is defined on X.
>
> Q - However, cost aspects of requiring more granular information for segmentation are not yet sufficiently addressed and discussed in my opinion - strengths and limits of the proposed method (if segmentation network is not well trained, then the approach won’t work)
>
> Ans - We discuss that the performance of Algorithm 2 depends on the semantic information obtained by the segmentation network  under the heading “Discussion” in section 4.1.3 on page 8. We agree that in comparison to the existing OOD detection methods, the proposed algorithms require more granular (pixel-level) information for removing the detection bias due to spurious background pixels. So, it is a tradeoff between the cost paid for extracting more granular information (ex. in training a good segmentation network on data sampled from in-distribution)  vs accuracy (low false OOD detection and high true OOD detection).

---

### Decision · Program_Chairs · 2023-01-20

**Decision:**

Reject

**Justification For Why Not Higher Score:**

Theory and method are disconnected, experimental comparison needs to be improved.

**Justification For Why Not Lower Score:**

N/A

**Metareview: Summary, Strengths And Weaknesses:**

Thanks to the authors for answering all questions. Nevertheless all reviewers kept their recommendation for rejection.

In general the reviewers appreciated the novel setting OOD detection where the authors argue that semantically similar (birds sitting on the ground) data should not be labeled as OOD even if the training data contains only birds sitting on trees. I would disagree here as in an application it might actually be very useful to detect inputs which are different from the training data (on which level is a question which depends on the application). Nevertheless, I agree that there might be applications where the discussed scenario is useful.

Currently, the theory section is overly complicated without much content. Theorem 1 is a trivial application of the triangle inequality and follows directly from the assumptions. However, the assumptions seem to have nothing to do with the story of the paper as the notion of semantically similar is not part of assumption 2) - the distribution D_I can have support anywhere in image space. The stated properties are also nowhere tested later on.

The comparison of OOD detectors based on pixelwise labeled data vs image-based label seem unfair. Moreover, the richer training data could be used to train the classifier in a way so that it depends less on the background which would be another fix to the problem.

The performance against near OOD data is not tested (other semantically similar classes) and the performance against SVHN as OOD data is significantly worse than the baselines.

In total the paper needs some revision before being ready for publication and the authors should integrate all the comments of the reviewers. I recommend to skip the theory section if it cannot be connected to the actual method.